# Mesenchymal Stromal Cells for Aging Cartilage Regeneration: A Review

**DOI:** 10.3390/ijms252312911

**Published:** 2024-11-30

**Authors:** Kun-Chi Wu, Yu-Hsun Chang, Dah-Ching Ding, Shinn-Zong Lin

**Affiliations:** 1Department of Orthopedics, Hualien Tzu Chi Hospital, Buddhist Tzu Chi Medical Foundation, Tzu Chi University, Hualien 970, Taiwan; drwukunchi@yahoo.com.tw; 2Department of Pediatrics, Hualien Tzu Chi Hospital, Buddhist Tzu Chi Medical Foundation, Tzu Chi University, Hualien 970, Taiwan; cyh0515@gmail.com; 3Department of Obstetrics and Gynecology, Hualien Tzu Chi Hospital, Buddhist Tzu Chi Medical Foundation, Tzu Chi University, Hualien 970, Taiwan; 4Institute of Medical Sciences, College of Medicine, Tzu Chi University, Hualien 970, Taiwan; 5Department of Neurosurgery, Hualien Tzu Chi Hospital, Buddhist Tzu Chi Medical Foundation, Tzu Chi University, Hualien 970, Taiwan

**Keywords:** mesenchymal stromal cells, cartilage regeneration, aging, osteoarthritis, tissue engineering, regenerative medicine

## Abstract

Cartilage degeneration is a key feature of aging and osteoarthritis, characterized by the progressive deterioration of joint function, pain, and limited mobility. Current treatments focus on symptom relief, not cartilage regeneration. Mesenchymal stromal cells (MSCs) offer a promising therapeutic option due to their capability to differentiate into chondrocytes, modulate inflammation, and promote tissue regeneration. This review explores the potential of MSCs for cartilage regeneration, examining their biological properties, action mechanisms, and applications in preclinical and clinical settings. MSCs derived from bone marrow, adipose tissue, and other sources can self-renew and differentiate into multiple cell types. In aging cartilage, they aid in tissue regeneration by secreting growth factors and cytokines that enhance repair and modulate immune responses. Recent preclinical studies show that MSCs can restore cartilage integrity, reduce inflammation, and improve joint function, although clinical translation remains challenging due to limitations such as cell viability, scalability, and regulatory concerns. Advancements in MSC delivery, including scaffold-based approaches and engineered exosomes, may improve therapeutic effectiveness. Potential risks, such as tumorigenicity and immune rejection, are also discussed, emphasizing the need for optimized treatment protocols and large-scale clinical trials to develop effective, minimally invasive therapies for cartilage regeneration.

## 1. Introduction

Cartilage degeneration is a common feature of both aging and osteoarthritis (OA), leading to impaired joint function [1]. OA is a major global health challenge, with increasing prevalence and burden worldwide. From 1990 to 2017, the global age-standardized prevalence of OA increased by 9.3% to 3754.2 per 100,000 [2]. By 2019, there were approximately 527.8 million prevalent cases globally [3]. Hip and knee OA are particularly significant, ranking as the 11th highest contributor to global disability in 2010 [4]. The burden of OA is expected to continue rising due to population growth, aging, and increasing obesity rates [5]. Socio-demographic factors play a role, with a positive association between age-standardized incidence rates and the Socio-demographic Index [3]. To address this growing burden, it is crucial to implement measures targeting risk factors, such as high body mass index, and to improve awareness among populations and policymakers about OA management [2]. In Taiwan, the overall crude incidence of total knee replacement rose from 26.4 to 74.55 per 100,000 persons between 1996 and 2010 [6].

In healthy joints, articular cartilage provides a smooth, resilient surface that absorbs mechanical stress and facilitates movement [7]. However, with aging, the ability of chondrocytes to maintain the balance between cartilage synthesis and degradation declines [8]. This reduction results in decreased production of essential extracellular matrix (ECM) components, such as collagen and proteoglycans, leading to a loss of cartilage structural integrity, thickness, and elasticity [9]. Aged chondrocytes also exhibit signs of senescence, producing inflammatory molecules that further inhibit cartilage regeneration [7].

Currently, OA management primarily focuses on symptom relief, joint function improvement, and quality of life enhancement, as no treatments are available to reverse cartilage degeneration [10]. Pain management is essential to OA treatment and often begins with non-pharmacological interventions, such as weight loss, physical therapy, and exercise, which help reduce joint stress and improve mobility [11]. Pharmacological options include acetaminophen and nonsteroidal anti-inflammatory drugs to manage pain and inflammation [12]. In more severe cases, intra-articular injections, such as corticosteroids or hyaluronic acid, may provide temporary relief [13]. However, these treatments do not halt disease progression. When conservative therapies fail and OA severely affects daily life, total joint replacement, particularly of the knee or hip, is often the preferred intervention [14]. Joint replacement surgery effectively alleviates pain and restores function but is typically reserved for end-stage OA due to its invasive nature and the potential need for revision surgeries over time [15].

Mesenchymal stromal cells (MSCs) have emerged as a promising therapeutic option for cartilage regeneration due to their unique ability to differentiate into various cell types, including chondrocytes, which are responsible for maintaining healthy cartilage [16]. Derived from tissues, such as bone marrow, adipose tissue, and umbilical cord blood, MSCs possess the ability for self-renewal and multipotency, making them ideal candidates for repairing damaged cartilage [17,18]. In addition to their differentiation potential, MSCs exhibit strong immunomodulatory and anti-inflammatory properties [19], which help reduce inflammation and support tissue repair in osteoarthritic joints [20]. In contrast to conventional treatments that focus on symptom management, MSC-based therapies address the root cause of cartilage degeneration by promoting tissue regeneration, restoring joint function, and potentially slowing or reversing disease progression [21]. With encouraging results in preclinical studies, MSCs hold the potential to revolutionize the treatment of OA and age-related cartilage deterioration, although challenges remain in translating these findings into effective clinical applications [22].

The aim of this review is to provide a comprehensive analysis of the potential role of MSCs and their derivatives in regenerating aging and osteoarthritic cartilage. Given that cartilage degeneration is a significant challenge in aging populations and OA, with traditional treatments focusing primarily on symptom management rather than on reversing tissue damage, this review explores MSC-based therapies as a regenerative solution. The review highlights MSCs’ biological properties, mechanisms of action, and application in cartilage repair. It also examines the progress achieved in preclinical and clinical studies, discusses current challenges and limitations, and evaluates advanced delivery methods for optimizing MSC efficacy. The scope of the review extends to identifying future research needs and assessing the potential for MSC-based therapies to serve as viable alternatives for treating cartilage degeneration in aging and OA populations.

### Study Research Strategy

A systematic search was conducted using the keywords “mesenchymal stromal cells, aging, and cartilage” from their respective inceptions to 21 October 2024. Synonyms and related terms were also included to expand the scope. The bibliographies of relevant reviews and included studies were also examined. Table 1 provides an overview of the search strategy used for the PubMed database.

## 2. MSCs

### 2.1. Definition and Biological Properties

#### 2.1.1. Sources of MSCs (Figure 1)

MSCs can be derived from various tissues, each offering distinct advantages for therapeutic use. Bone marrow-derived MSCs (BM-MSCs) are the most extensively studied and show strong potential for differentiating into chondrocytes, making them a common choice for cartilage regeneration [23,24]. However, their collection is invasive, and both their quantity and differentiation capability tend to decrease with age [25]. Adipose tissue (AT)-derived MSCs offer a more abundant and accessible source, with high cell yields and regenerative potential comparable with bone marrow-derived MSCs [26]. Umbilical cord (UC)-derived MSCs have gained attention for their non-invasive collection, high proliferative capability, and low immunogenicity, making them suitable for allogeneic use [20]. Other potential sources include dental pulp, synovial membrane, and placental tissue, all of which contribute to the expanding range of options for MSC-based therapies [16]. The diversity of MSC sources provides flexibility for personalized treatments that meet specific clinical needs; however, variations in the properties of MSCs derived from different tissues remain an important consideration in therapeutic applications.

#### 2.1.2. Characteristics of MSCs

MSCs are characterized by their remarkable self-renewal and multipotent differentiation capabilities, which make them invaluable for regenerative medicine [16]. Self-renewal refers to the ability of MSCs to undergo multiple rounds of cell division while maintaining their undifferentiated state, ensuring a sustainable source of stromal cells for therapeutic applications [27]. This property is crucial for tissue regeneration, as it enables the expansion of MSCs in vitro before transplantation [28]. Multipotency, on the other hand, refers to the ability of MSCs to differentiate into various specialized cell types, including adipocytes (fat cells), osteoblasts (bone cells), and chondrocytes (cartilage cells) [18]. This versatility enables MSCs to contribute to the repair and regeneration of multiple tissues, particularly in the context of cartilage repair, where their ability to become chondrocytes supports the restoration of structural integrity in damaged cartilage [21]. Collectively, these characteristics make MSCs a promising solution for addressing various degenerative conditions, including OA and age-related cartilage degeneration.

The International Society for Cell and Gene Therapy (ISCT) minimal criteria for defining MSCs consist of three main components: plastic adherence, specific surface antigen expression, and multipotent differentiation potential [29]. The update ISCT recommendation includes using the term “mesenchymal stromal cells” for MSCs, reporting tissue of origin, and functional assays to test cells’ properties and potency [29,30].

#### 2.1.3. Comparing MSC Yields, Proliferative, and Differentiation Capacity from Different Sources

MSCs from different sources exhibit varying characteristics, impacting their potential for regenerative medicine applications. Studies comparing MSCs from BM, AT, UC, and decidua parietalis (DeP) reveal source-specific differences in yield, proliferation, and differentiation capacities. AT-MSCs demonstrated the highest isolation yield (BM-MSC-1 × 10^3^ cells/mL of bone marrow aspirate, AT-MSC-2.5 × 10^6^ cells/g of adipose tissue and AM-MSC-5.6 × 10^6^ cells/g of amniotic tissue) and proliferation rates (absorbance at 572 nm after 240 h, AT- 0.7, BM: 0.4, AT-0.2) [31], while UC-MSCs showed superior proliferation (doubling time: 17.7 vs. 21.9 h) and colony formation (16% vs. 13.6%) compared to DeP-MSCs [32]. BM-MSCs exhibited better differentiation abilities, making them preferable for orthopedic applications [33]. Immunophenotyping confirmed typical MSC surface markers across sources, with slight variations in quantitative data between laboratories [34]. Functional assays revealed source-specific differences in angiogenic and immunomodulatory properties, with BM-MSCs enhancing tubulogenesis and AT-MSCs showing superior immunosuppressive abilities [34]. Dental pulp (DP)-MSCs demonstrate superior osteogenic differentiation potential and lower apoptosis rates compared to UC-MSCs, but UC-MSCs exhibit higher proliferation capacity [35]. While DP-derived cells show higher colony-forming efficiency, BM-MSCs have greater expansion success and differentiation potential [36]. UC-MSCs express higher levels of MSC surface markers like CD29, CD34, CD44, CD73, CD105, CD146, and CD166 compared to DP, although both sources exhibit similar overall MSC marker expression [37]. Gene expression patterns differ between UC and DP, with UC showing higher expression of cell proliferation and angiogenesis-related genes, while DP expresses more growth factor and signal transduction-related genes [37]. These findings highlight the importance of considering tissue origin when selecting MSCs for specific clinical applications.

**Figure 1 ijms-25-12911-f001:**
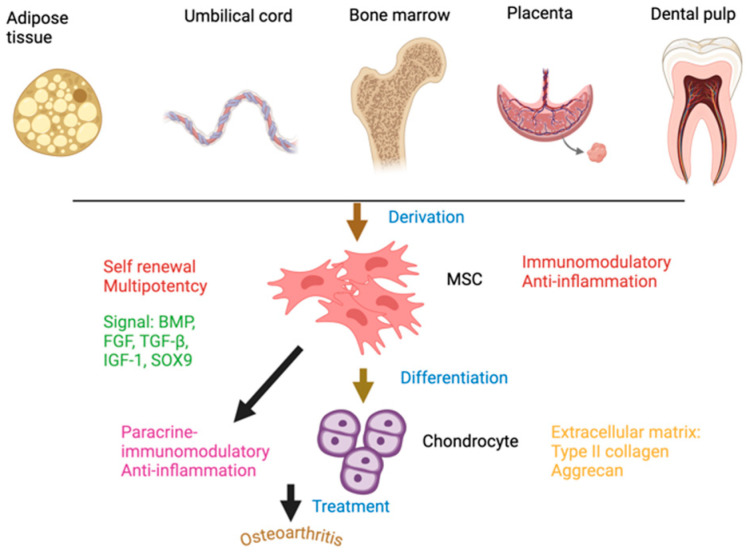
Sources, properties, and therapeutic potential of mesenchymal stromal cells (MSCs) for osteoarthritis treatment. MSCs can be derived from adipose tissue, umbilical cord, bone marrow, placenta, and dental pulp. MSCs from these sources exhibit essential characteristics, including self-renewal and multipotency, and respond to specific signaling molecules (e.g., BMP, FGF, TGF-β, IGF-1, SOX9) that aid in their proliferation and differentiation. MSCs also possess immunomodulatory and anti-inflammatory properties, making them suitable for therapeutic applications. Upon differentiation, MSCs can become chondrocytes, cells critical for cartilage formation and maintenance. Chondrocytes produce key ECM components, including type II collagen and aggrecan, which are essential for joint health. The paracrine and immunomodulatory effects of MSCs, along with their differentiation potential, are utilized in treating osteoarthritis to alleviate inflammation, support tissue repair, and restore cartilage function. This approach offers a promising approach for managing joint degenerative diseases.

#### 2.1.4. Immunomodulatory and Anti-Inflammatory Properties of MSCs

MSCs possess significant immunomodulatory and anti-inflammatory properties, enhancing their therapeutic potential for treating degenerative diseases such as OA [19]. These cells modulate immune responses through secretion of various bioactive factors, including cytokines, growth factors, and extracellular vesicles (EVs), which help create a favorable microenvironment for tissue repair [38,39]. MSCs exhibit strong immunosuppressive properties mediated by HLA-G expression, which can inhibit the proliferation of peripheral blood mononuclear cells and the expression of inflammatory cytokines [19]. MSCs inhibit the activation and proliferation of immune cells, such as T cells and natural killer cells, reducing inflammation and preventing excessive tissue damage [40,41]. They also promote macrophage polarization from a pro-inflammatory (M1) to an anti-inflammatory (M2) phenotype, further mitigating inflammatory responses [42,43]. Additionally, MSCs secrete factors, such as PGE-2, CCL-2, and micro-RNAs, that promote the recruitment of regulatory immune cells, regulating the polarization, migration, and function of macrophages and enhancing their ability to control inflammation [44]. This immunomodulatory effect not only helps alleviate the symptoms of OA but also supports healing by mitigating the adverse effects of chronic inflammation on cartilage and surrounding tissues. As a result, MSCs are considered a promising candidate for regenerative therapies targeting joint disorders [45].

### 2.2. Mechanisms of Cartilage Regeneration

#### 2.2.1. MSC Differentiation into Chondrocytes

MSCs have the remarkable ability to differentiate into chondrocytes, the specialized cells responsible for maintaining cartilage tissue [17]. This differentiation process is crucial for cartilage regeneration, as MSCs can be induced to adopt the chondrocyte phenotype under specific conditions, such as exposure to growth factors like transforming growth factor-beta (TGF-β) and bone morphogenetic proteins (BMPs) [46,47]. Once differentiated, MSC-derived chondrocytes produce essential ECM components, including collagen type II and aggrecan, which are crucial for the structural integrity and functionality of healthy cartilage [48,49]. Furthermore, the joint microenvironment, along with scaffold materials or biomimetic surfaces, enhances the successful differentiation of MSCs into chondrocytes [50]. This process not only replenishes damaged cartilage but also restores its biomechanical properties, offering a potential therapeutic approach for degenerative conditions like OA. By directly contributing to cartilage tissue regeneration, MSC differentiation into chondrocytes represents a promising avenue for long-term joint repair and functional recovery [51].

One of the argued issues is lifespan of transplanted cells. A study found no transplanted cells stained after 2 months of treatment [52]. Other research indicates that MSCs have a limited lifespan [53]. Injecting MSCs in gel form into the articular cavity has demonstrated improved patient outcomes [54], though implanting MSCs directly into cartilage defects under arthroscopy has shown even greater benefits [55]. In an animal study, bone marrow-derived MSCs cultured and implanted with a collagen-hyaluronic acid scaffold significantly enhanced type II collagen [56]. The therapeutic effect of MSCs direct injections need further exploration.

Another issue is stem cell’s homing effect. The homing effect of MSCs to injured joints is a crucial factor in their therapeutic efficacy [21]. However, challenges remain in ensuring MSC retention and engraftment in cartilage tissue. To address this, researchers have explored various strategies, including cell surface modification and the use of nanoparticles for improved targeting and gene delivery [57]. Additionally, advanced biomaterials have been investigated to enhance MSC engraftment to cartilage and optimize cell dosage [57]. These approaches aim to improve the overall efficacy of MSC-based therapies for OA treatment.

MSC therapy shows promise for treating knee OA, with both autologous and allogeneic sources demonstrating potential benefits. Allogeneic MSCs have shown improvements in pain, function, and cartilage quality compared to hyaluronic acid injections [58]. While both sources appear safe, autologous MSCs may offer superior efficacy and safety profiles [59]. However, allogeneic MSCs provide logistical advantages and consistent product quality [60]. Despite these findings, current evidence is limited, and more high-quality randomized controlled trials comparing autologous and allogeneic MSCs are needed to establish definitive recommendations for treating knee OA [59,60].

Transplanted cell numbers is also matter. Cell doses ranging from 24 to 100 million have been investigated, with moderate doses (40 million) appearing optimal for efficacy and safety [61,62]. Lamo-Espinosa et al. reported that the same beneficial effects of MSC treatment with different cell dosages (10, 40 and 100 million cells) can persist for up to 4 years after a single injection [63]. Additionally, the use of adipose-derived stromal vascular fraction has shown potential in improving knee function and reducing pain, with higher cell numbers (an average of 45 million) correlating with better outcomes [64]. While these studies demonstrate the safety and efficacy of MSC-based treatments for knee osteoarthritis, larger-scale, long-term clinical trials are needed to further validate these findings [61].

#### 2.2.2. Paracrine Signaling

Paracrine signaling plays a pivotal role in the therapeutic effects of MSCs, particularly in cartilage regeneration [65]. Rather than relying solely on direct differentiation into chondrocytes, MSCs exert much of their regenerative potential by secreting a variety of bioactive molecules, including growth factors, cytokines, and EVs [66]. These molecules are released into the surrounding microenvironment, where they influence nearby cells and tissues. Growth factors, such as TGF-β, fibroblast growth factor (FGF), and insulin-like growth factor (IGF), promote chondrocyte proliferation and enhance the production of key ECM components, such as collagen and proteoglycans, which are essential for cartilage repair [67,68]. Additionally, cytokines secreted by MSCs modulate inflammation by reducing pro-inflammatory mediators and promoting an anti-inflammatory environment conducive to healing [38]. In inflammation-primed MSCs, various cytokines (e.g., CXCL9, 5, 2, and 7) attract immune cells, while inducible nitric oxide synthase and indoleamine 2,3-dioxygenase are produced to suppress T-cell activity [38]. Through these paracrine actions, MSCs enhance tissue regeneration, protect existing cartilage from further damage, and create a favorable environment for joint repair, making paracrine signaling a critical mechanism in MSC-based therapies for OA and cartilage degeneration [69].

Glucose plays a crucial role in maintaining MSC survival and function in the paracrine environment. Studies have shown that glucose deprivation, more than oxygen shortage, severely compromises MSC viability and functional maturation [70,71]. Under near-anoxic conditions, MSCs rely almost exclusively on glucose for ATP production through anaerobic glycolysis and possess limited internal glucose reserves [72]. Glucose not only promotes MSC survival but also enhances their angiogenic potential [71]. To address this challenge, researchers have developed glucose-releasing scaffolds and enzyme-controlled, nutritive hydrogels that provide physiological glucose levels to MSCs [73]. These approaches have demonstrated improved MSC viability and paracrine functions both in vitro and in vivo. The findings highlight the importance of glucose in the paracrine environment for maintaining MSC survival and suggest that glucose supplementation strategies could enhance the efficacy of MSC-based therapies.

#### 2.2.3. Immunomodulatory and Anti-Inflammatory Effects

MSCs possess potent immunomodulatory and anti-inflammatory effects, making them a valuable therapeutic option for inflammatory conditions like OA [74]. MSCs regulate immune responses by interacting with various immune cells, including T cells, B cells, natural killer cells, and macrophages [41]. Through the release of cytokines and growth factors, MSCs suppress the activity of pro-inflammatory T helper (Th1) cells and enhance the generation of regulatory T cells (Tregs), promoting an anti-inflammatory state [75]. Additionally, MSCs induce a shift in macrophages from a pro-inflammatory (M1) to an anti-inflammatory (M2) phenotype, further reducing joint inflammation [76]. These anti-inflammatory actions help mitigate the chronic inflammation typically associated with OA, thereby protecting cartilage from further degradation and creating a more conducive environment for tissue repair [77,78]. By modulating immune activity and reducing inflammation, MSCs not only alleviate OA symptoms but also support long-term cartilage regeneration and joint health.

## 3. MSC-Based Therapies for Cartilage Repair

### 3.1. Preclinical Studies

#### Animal Models Demonstrating MSC-Induced Cartilage Regeneration

Animal models have been instrumental in demonstrating the potential of MSCs for cartilage regeneration [22]. Preclinical studies have shown that MSCs significantly enhance cartilage repair in various species, including mice [17], rats [79], rabbits [18], and larger animals, such as pigs [80], sheep [81], and horses [82]. These models mimic the degenerative conditions of OA or cartilage injury observed in humans, enabling researchers to study the effects of MSC therapies [83]. Animal studies have reported improvements in joint function, reductions in cartilage degradation, and restoration of cartilage structure, with increased levels of collagen type II and proteoglycans [18]. These findings provide strong evidence for the therapeutic potential of MSCs in cartilage repair, establishing a foundation for clinical trials targeting OA and cartilage injuries in humans [80,83].

In various OA animal models, rats were commonly used, with studies reporting improved cartilage preservation and reduced damage after MSC injections [84,85]. In rabbits, most studies found improvements in cartilage repair and reduced osteoarthritis progression based on histological and radiological assessments [86]. Mice models showed promising results in fracture-induced and collagenase-induced OA [87]. Interestingly, a study on Guinea pigs with spontaneously developing OA reported beneficial effects of MSC treatment [88]. Larger animals provided mixed results. In sheep and goat models, some studies reported improvements, while others found no significant differences between MSC-treated and control groups [89,90]. A study using horses found no significant treatment effects on gross pathological observations [91]. Several studies using pigs found significant treatment effects on gross pathological observations [80,83]. Overall, while most studies across species reported some degree of improvement after MSC treatment, the outcomes varied considerably [92]. Smaller animals generally showed more consistent positive results, while larger animal models produced more mixed outcomes. However, it’s important to note that the review found substantial heterogeneity among studies in terms of methodologies, MSC types, doses, and evaluation methods [84]. Additionally, the evidence quality for all outcomes was either low or very low, highlighting the need for further high-quality research before drawing definitive conclusions about the efficacy of MSC treatments across species [84].

### 3.2. Clinical Trials

#### 3.2.1. Key Findings from Human Trials

MSCs have shown promise in cartilage repair and OA treatment. Clinical trials have demonstrated the safety and potential efficacy of MSC-based therapies, with significant pain reduction reported in treated patients compared with controls [93]. Various MSC sources, including umbilical cord blood and autologous or allogeneic cells, have been investigated [94,95]. Intra-articular MSC injections have shown encouraging results in cartilage regeneration, although achieving complete repair of articular cartilage defects remains challenging [94]. Long-term follow-up studies have reported durable cartilage regeneration and improved clinical outcomes without significant adverse events [96]. However, the heterogeneity of current clinical trials highlights the need for more robust and consistent studies to generate reliable evidence supporting MSC therapies for OA [94,97]. Future research directions include investigating MSC-derived EVs, miRNAs, and advanced gene-editing techniques [98].

We conducted a search using “mesenchymal stromal cells” and “osteoarthritis” as keywords to identify relevant clinical trials listed on clinicaltrials.gov. A total of 130 trials were conducted up to October 28, 2024. Of these, 35 studies are currently open, while the remaining are either closed or have an unknown status (Table 2). Among transplanted cell types, AT-MSCs were the most commonly used, followed by BM-MSCs and UC-MSCs (Table 3). Autologous cells are the primary sources used for transplantation (Table 3).

#### 3.2.2. Challenges in Translating Preclinical Success to Clinical Practice

MSCs show promise for treating OA and other conditions, but translating preclinical success to clinical practice involves significant challenges. These challenges include the heterogeneity of MSC populations, variability in isolation and expansion protocols, and concerns about cell viability and function after delivery [99,100].

MSCs exhibit significant heterogeneity in morphology, phenotype, and function, which poses challenges for their therapeutic applications [101,102]. This heterogeneity is influenced by microenvironmental factors, including mechanical stiffness, which can alter MSC gene expression and commitment [103]. To address this variability, researchers have explored marker-based isolation strategies and high-throughput approaches to identify and purify MSC subpopulations [101]. Studies have shown that culturing MSCs on soft surfaces or inhibiting specific pathways can promote a more homogeneous population [101]. Interestingly, while individual MSC clones may display varying immunosuppressive capabilities, exposure to pro-inflammatory cytokines (licensing) can eliminate these functional differences, resulting in uniformly enhanced immunosuppressive activity mediated by factors such as nitric oxide and prostaglandin E2 [104].

Recent studies highlight the variability in MSC isolation and expansion protocols, emphasizing the need for standardization. Todtenhaupt et al. developed a robust method for human umbilical cord-derived MSCs, optimizing critical variables and demonstrating consistency across 90 donors [105]. Rojewski et al. successfully translated a standardized protocol for bone marrow-derived MSCs from validation to clinical manufacturing, showing stable performance characteristics despite variations in starting material [106]. Shaz et al. found that local manufacturing processes contribute to variability in MSC expansion, while growth media supplements affect gene expression and cell function [107]. Wright et al. optimized protocols for canine umbilical cord-derived MSCs, addressing the challenge of maintaining these cells in culture for extended periods [108]. Their method improved cellular adherence, colony-forming efficiency, and population doubling times. These studies collectively emphasize the importance of developing standardized, robust protocols to enhance reliability and comparability of results across different donors and studies.

MSCs show promise for cellular therapies, but maintaining their viability during storage and transplantation is crucial. Cell viability should be at least 80% for clinical use [109]. However, MSC viability decreases rapidly after dissociation from culture dishes [110]. Storage solutions significantly affect cell survival, with viability dropping below 70% after 6 h in common parenteral solutions [111]. Factors influencing MSC viability include cell density, dimethylsulfoxide (DMSO) concentration, and needle gauge. Maintaining cell density below 2 × 10^7^ cells/mL and DMSO concentration below 0.5% can help preserve viability above 82% when using 25- or 27-gauge needles [112]. Various analytical techniques, such as membrane integrity assays, morphological studies, and fluorescence biosensors, are used to assess MSC viability [109]. Optimizing storage conditions and transplantation methods is crucial for maintaining MSC viability and therapeutic potential.

MSCs show promise in improving outcomes after organ transplantation. Studies have demonstrated the safety of MSC infusion after liver transplantation [113] and kidney transplantation [114]. MSCs have shown potential in treating poor graft function following hematopoietic cell transplantation, with improved hematological responses and reduced transfusion requirements [115]. However, the therapeutic effects of MSCs may be limited due to impaired function after infusion. Preconditioning methods are being explored to enhance MSC efficacy in kidney transplantation [116]. While MSC therapy appears safe and feasible, its benefits in organ transplantation are not yet fully established. Further research is needed to optimize MSC function post-transplantation and to demonstrate their potential advantages over standard immunosuppressive regimens. Larger prospective studies are required to confirm the efficacy of MSC therapy in transplantation settings [113,114].

Optimizing cell delivery methods, addressing hemocompatibility issues, and ensuring safety are crucial for successful clinical translation [100]. While MSCs demonstrate cartilage repair potential and immunomodulatory properties in preclinical studies, understanding cell behavior post-transplantation and enhancing potency within disease microenvironments remain important goals [117].

An alternative to whole-cell therapy, MSC-derived EVs offer potential advantages, including a higher safety profile and lower immunogenicity. However, EV research is confronted with challenges in production methods, characterization, pharmacokinetics, and safety assessments, which need to be addressed before clinical application can proceed [118].

Endochondral ossification (EO) is a crucial process in bone formation and repair, involving the transformation of cartilage into bone [119]. Recent research challenges the traditional view that hypertrophic chondrocytes undergo apoptosis, suggesting instead that they may transdifferentiate into osteoblasts [119,120]. This chondrocyte-to-osteoblast transdifferentiation occurs through three models: direct transdifferentiation, dedifferentiation to redifferentiation, and chondrocyte to osteogenic precursor [120]. Epigenetic factors, including DNA methylation, histone modifications, and non-coding RNAs, play crucial roles in regulating EO and chondrogenesis [121]. Understanding these processes is essential for developing treatments for skeletal diseases and OA, which involves disruptions in chondrocyte homeostasis [122]. Maintaining healthy articular cartilage is vital for joint function and longevity, and further research into cartilage development and homeostasis is necessary for establishing regenerative therapies [122].

## 4. Advancements in MSC Delivery Methods

### 4.1. Biocompatible Scaffolds for Better MSC Retention and Differentiation

Recent studies have explored the use of biocompatible scaffolds to enhance MSC retention and chondrogenic differentiation [123]. A novel scaffold composed of tricalcium phosphate, collagen, and hyaluronate has demonstrated superior chondroinductive properties compared with scaffolds without hyaluronate [124]. Additionally, gelatin scaffolds with aligned pore architecture have been shown to improve cellular infiltration, alignment, and chondrogenic differentiation of infrapatellar fat pad-derived MSCs compared with those with random pore structures [125]. Electrospun gelatin/glycosaminoglycan nanofiber scaffolds, particularly those containing 15% glycosaminoglycan, have enhanced chondrogenic differentiation of bone marrow-derived MSCs [126] Additionally, poly(l-lactide-co-glycolide)/poly(l-lactide) microfiber scaffolds coated with human fibroblast-derived matrix and immobilized with TGF-β1 have promoted MSC condensation and improved chondrogenesis both in vitro and in vivo [127]. These studies highlight the importance of scaffold composition and architecture in promoting MSC retention and chondrogenic differentiation for cartilage tissue engineering applications.

### 4.2. Cell-Free Therapies

#### 4.2.1. Engineered EVs for Delivering MSC-Derived Factors

EVs are small, membrane-enclosed structures released by cells into the extracellular space, ranging in size from 30 nm to 5 μm [128]. They play crucial roles in cell-to-cell communication by transferring functional biomolecules between cells [129]. EVs are classified into several subtypes based on their biogenesis, including exosomes (40–100 nm), microvesicles (100–1000 nm), and apoptotic bodies [130]. Recent studies have identified additional EV types such as autophagic EVs, stressed EVs, and matrix vesicles [130]. EVs are involved in various physiological and pathological processes, including tumor immunosuppression and metastasis [130]. They show promise as biomarkers, therapeutic agents, and drug delivery vehicles [131]. However, the cellular and molecular mechanisms governing EV functions are not fully understood, partly due to technical challenges in studying these small particles [131]. Despite these hurdles, the versatility and potential of EVs in regenerative medicine make them an exciting area of research for future therapeutic applications [132].

Engineered EVs have emerged as a novel approach for delivering MSC-derived factors to enhance cartilage regeneration [133]. These vesicles, including exosomes, are naturally secreted by MSCs and carry bioactive molecules, such as proteins, lipids, and RNAs, that facilitate intercellular communication and tissue repair [134]. Engineering EVs enables the optimization of their cargo to deliver specific therapeutic agents, such as growth factors, anti-inflammatory cytokines, and microRNAs, which promote chondrocyte proliferation and ECM production [135,136]. By customizing the content and surface properties of EVs, researchers can enhance their stability, targeting capabilities, and therapeutic efficacy [137]. These engineered vesicles provide a cell-free, minimally invasive alternative to traditional MSC therapies, avoiding potential complications associated with cell transplantation while retaining regenerative benefits [138]. Moreover, EVs exhibit low immunogenicity, making them suitable for repeated administration and potentially for allogeneic use [139,140]. This emerging strategy represents a promising advancement in MSC-based therapies, offering a precise and efficient method for promoting cartilage repair and mitigating OA progression.

#### 4.2.2. EVs as a Promising Therapeutic Tool

Mesenchymal stromal cell-derived EVs (MSC-EVs) present a promising alternative to cell-based therapies for tissue regeneration and disease treatment [141]. As nano-sized lipid structures, MSC-EVs facilitate cell–cell communication by transporting bioactive molecules between cells [142]. MSC-EVs offer several advantages over cell therapies, including lower immunogenicity, improved barrier crossing, and fewer safety concerns [141]. However, challenges, such as low production, poor retention, and limited targeting, hinder their clinical application [143]. To address these issues, researchers are exploring bioengineering strategies to enhance the therapeutic potential of MSC-EVs. These strategies include cargo and surface modifications to improve targeting and efficacy [142,143]. Additionally, novel delivery systems, such as biodegradable hydrogels, are being developed to overcome rapid biodegradation and clearance of EVs in vivo [144]. These advancements aim to maximize the therapeutic potential of MSC-EVs across various applications in regenerative medicine and tissue engineering.

### 4.3. Gene Editing and Bioengineering Approaches to Enhance MSC Effectiveness

Recent research has focused on enhancing MSC effectiveness in chondrogenesis to promote cartilage regeneration. Various strategies have been explored, including the optimization of bioactive factors, culture conditions, cell type selection, coculture techniques, gene editing, scaffold development, and physical stimulation [145]. Gene editing techniques, particularly CRISPR/Cas9, have shown promise in modifying MSCs to improve their chondrogenic potential [146,147,148]. CRISPR/Cas9 has been used to knockout genes like VEGF and RUNX2 in human MSC lines, resulting in engineered extracellular matrices (eECMs) with improved cartilage repair capabilities [149]. Efficient RUNX2 knockdown using CRISPR-Cas9 has been shown to decrease osteogenic differentiation and increase chondrogenic differentiation of MSCs [150]. This approach has potential for treating rheumatoid arthritis and repairing chondral lesions [146]. While CRISPR/Cas9 offers precise gene targeting, further research is needed to optimize MSC-based therapies for clinical applications [151]. Overall, CRISPR/Cas9 editing of MSCs shows promise for enhancing cartilage regeneration and developing tailored eECMs for tissue engineering applications.

Genetically engineered MSCs may offer new treatment options for arthritic joints and chondral lesions. Additionally, bottom-up strategies, such as gene delivery, gene editing, and subpopulation isolation, have been investigated to enhance MSC behavior for cartilage tissue engineering [152,153]. Despite progress, challenges remain in achieving durable and phenotypically accurate regenerated cartilage. Continued research in MSC engineering and chondrogenic differentiation is crucial for advancing cartilage tissue engineering and developing effective treatments for joint diseases [145,146].

CRISPR/Cas9 gene editing has shown promise for treating diseases, but off-target effects remain a significant concern [154,155]. These unintended alterations can include small indels, large deletions, and structural variations, posing risks to patients [155]. Various methods have been developed to detect and assess off-target effects, with ongoing efforts to enhance CRISPR precision [154]. Population-specific analysis of potential off-target sites is crucial, as polymorphisms can affect cleavage likelihood and create new PAM sequences [156]. Additionally, complex on-target outcomes, such as large deletions and gene rearrangements, have been observed and require careful consideration [157]. As CRISPR-based therapies advance towards clinical applications, comprehensive evaluation of both off-target and on-target effects is essential to ensure safety and efficacy in gene therapy development [155,157].

## 5. Future Directions and Research Needs

### 5.1. Large-Scale Clinical Trials and Long-Term Outcomes

Recent studies have shown promising results for MSC therapy in treating OA. A 7-year follow-up study demonstrated long-term clinical benefits and improved cartilage structure after a single injection of adipose-derived MSCs in knee OA patients [158]. Intra-articular injections of allogeneic bone marrow MSCs showed significant improvements in pain, function, and cartilage quality compared to hyaluronic acid [58]. Similarly, autologous bone marrow MSCs exhibited long-term clinical benefits, with both low (10 × 10^6^) and high (100 × 10^6^) doses outperforming hyaluronic acid in pain reduction and functional improvement [159,160]. A dose-escalation study using umbilical cord-derived MSCs found that all tested doses (2 × 10^6^, 20 × 10^6^, and 80 × 10^6^) were safe and effective, with medium and low doses showing superior outcomes [161]. These studies consistently report no serious adverse events. Overall, these findings indicate that MSC therapy may offer promising long-term benefits for OA treatment, with multiple injections potentially yielding better results.

However, large-scale clinical trials and long-term outcomes are critical for fully understanding the therapeutic potential of MSCs in cartilage repair and their application in treating conditions such as OA [21]. While early-phase trials have shown promising results in terms of safety and short-term efficacy, large-scale, randomized controlled trials are necessary to validate these findings and provide robust data on the long-term effectiveness of MSC therapies [162]. Key outcomes, such as sustained cartilage regeneration, functional improvements, and joint preservation, remain relatively understudied [163]. Future research needs to address challenges, including optimizing MSC source, dosage, and delivery methods, as well as understanding patient-specific factors that may influence treatment outcomes [163]. Additionally, exploring the use of MSC-derived products, such as EVs or engineered scaffolds, may offer alternative or complementary strategies for cartilage repair [164,165]. Establishing standardized protocols for MSC preparation and administration, along with long-term follow-up, will be essential for translating these therapies into routine clinical practice [166]. Ongoing and future large-scale trials will provide critical insights into the durability of MSC-induced cartilage regeneration and help determine their role in managing OA and cartilage injuries [167].

### 5.2. Optimization of Delivery Methods

Recent studies have investigated the efficacy of intra-articular injections of MSCs for knee OA treatment. Clinical trials have shown improvements in pain, function, and quality of life following MSC injections [168,169,170]. Magnetic resonance imaging revealed decreased cartilage defects and increased cartilage volume in treated knees [168,169]. Histological analysis demonstrated hyaline-like cartilage regeneration [169]. Higher doses of MSCs (1.0 × 10^8^ cells) generally produced better outcomes [168,169]. However, concerns about the durability of clinical and structural improvements beyond one year were noted, particularly for lower doses [168]. No serious adverse events were reported in these studies [168,169,170]. While promising, larger randomized clinical trials with long-term follow-up are needed to establish the efficacy and optimal dosing of MSC injections for knee OA [61].

Optimizing delivery methods is crucial for enhancing the effectiveness of MSCs in cartilage repair (Figure 2). Various strategies, including the use of scaffolds, exosomes, and hydrogels, have been explored to improve MSC retention, survival, and functionality at the injury site [171,172]. Scaffolds, typically made from biocompatible materials, provide a structural framework that mimics the cartilage matrix, supporting MSC differentiation into chondrocytes and promoting ECM production [123,173]. By contrast, hydrogels offer a flexible and dynamic environment that encapsulates MSCs, protecting them from degradation while delivering them precisely to damaged cartilage [174]. These hydrogels can be engineered for gradual release of MSCs or their secreted factors, thereby enhancing tissue repair over time [175]. Exosomes, small vesicles secreted by MSCs, are emerging as a promising cell-free alternative, capable of delivering therapeutic molecules like proteins and RNAs to promote cartilage regeneration [176]. Combining these delivery systems with MSCs or their derivatives holds great potential for maximizing regenerative outcomes and improving clinical results in cartilage repair therapies.

### 5.3. Investigating Combination Therapies

Human umbilical cord MSCs combined with hyaluronic acid showed synergistic effects in ameliorating knee OA progression in rats [177]. A meta-analysis revealed that MSCs combined with platelet-rich plasma (PRP) improved pain and joint function in OA patients compared to controls or hyaluronic acid alone [178]. MSCs have shown promise in OA treatment due to their differentiation potential and immunomodulatory effects, though further research is needed to evaluate safety and effectiveness [179]. In vitro and in vivo studies demonstrated that combining adipose-derived MSCs with chondrocytes reduced oxidative stress-induced damage, improved cartilage repair, and enhanced expression of cartilage-specific genes compared to either cell type alone [180]. These findings suggest that combination therapies involving MSCs may offer improved outcomes for OA treatment.

Investigating combination therapies for OA treatment, such as MSCs combined with growth factors or drugs, offers a promising approach to enhance tissue regeneration and reduce inflammation. When combined with growth factors, like TGF-β or BMPs, the regenerative effects of MSCs are amplified, promoting cartilage repair and reducing joint degradation [181,182]. Additionally, combining MSCs with drugs, such as anti-inflammatory agents or disease-modifying osteoarthritis drugs like kartogenin, can help control the inflammatory environment within the joint, further improving patient outcomes [183,184]. The combination of placental-derived MSCs and stigmasterol demonstrated enhanced cartilage repair and regeneration in a rat OA model [185]. Similarly, adipose-derived MSCs combined with chondrocytes reduced oxidative stress-induced damage and improved cartilage formation in vitro and in vivo [180]. Another study found that combining tissue inhibitors of metalloproteinase-3, sulfated carboxymethylcellulose, and piperlongumine synergistically reduced inflammation, cartilage matrix loss, chondrosenescence, and oxidative stress in ex vivo OA models [186]. Although these combination therapies remain under active investigation, they represent a multi-targeted strategy to address both the symptoms and underlying pathophysiology of OA.

### 5.4. The Value of Magnetic Resonance Imaging for Sequential Non-Invasive Clinical Monitoring

Magnetic resonance imaging (MRI) has emerged as a valuable tool for non-invasive monitoring of stem cell therapy in OA. Studies have shown that MRI can effectively evaluate cartilage regeneration and treatment outcomes [187]. T1 and T2 mapping techniques have been used to assess cartilage quality and correlate with clinical scores [187]. Pre-treatment MRI findings, such as cartilage lesions and bone marrow lesions, have been associated with improved clinical outcomes following adipose-derived stem cell therapy [188]. Advanced imaging techniques, including diffusion-weighted MRI and contrast-enhanced ultrasound, show promise for stem cell treatment evaluation [189]. In a study using Wharton’s jelly mesenchymal stromal cells for knee OA, MRI scans demonstrated significant improvements in various parameters, including cartilage loss and bone marrow lesions, correlating with functional improvements [190]. These findings highlight the potential of MRI as a crucial tool for monitoring stem cell therapy efficacy in OA.

### 5.5. Potential for Personalized Regenerative Medicine Approaches

Personalized regenerative medicine using MSCs for OA holds significant potential because it provides treatments tailored to individual’s needs [191]. MSCs have a distinctive ability to regenerate damaged cartilage, modulate inflammation, and adapt to the specific biological environment of a patient’s joint [69]. By analyzing patient-specific factors, such as the severity of cartilage damage, inflammation levels, and genetic predispositions, treatments can be personalized to enhance efficacy [192]. Biomarkers play a crucial role in identifying early-stage OA and predicting disease progression, enabling more effective personalized treatments [193]. The concept of endotypes and phenotypes in OA helps explain variations in clinical manifestations, etiology, and pathophysiology, potentially improving patient selection for regenerative therapies [194]. While regenerative medicine initially focused on cell engraftment and differentiation, recent studies suggest that injected cells may primarily exert transient paracrine effects before undergoing apoptosis [195]. To justify the high cost of cell therapy, long-term cell survival and durable structural improvements are essential [195,196]. Understanding OA phenotypes and endotypes can contribute to more effective patient selection for regenerative treatments, potentially reducing healthcare costs and improving outcomes [194]. This personalized approach enables for optimization of MSC source (whether autologous or allogeneic), dosage, and potential combination with other therapies, such as growth factors or drugs, ensuring that the therapy is best suited to the individual’s condition [197,198]. As research advances, personalized MSC therapies could transform OA management by providing more effective, long-lasting, and targeted treatments, potentially reducing the need for invasive procedures like joint replacement [94].

### 5.6. Innovative Approaches in Cell Engineering

Recent advances in MSC engineering for cartilage regeneration focus on enhancing chondrogenic differentiation and addressing hypertrophy. Three-dimensional cell sheet technology has emerged as a promising approach for fabricating hyaline-like cartilage constructs [199]. Multilayering MSC sheets can increase construct thickness and enhance cellular interactions, although an optimal thickness threshold exists for maximizing chondrogenesis [199]. Various strategies to improve MSC chondrogenesis include optimizing bioactive factors, culture conditions, and physical stimulation [145]. Notably, N-cadherin mimetic hydrogels have been shown to attenuate hypertrophy and enhance chondrogenesis by regulating cell metabolism, specifically glycolysis and fatty acid oxidation [200]. Mechanobiological stimulation of stem cells has also been investigated to recreate specific environmental conditions for chondrogenesis [201]. Hydrogels, both natural and synthetic, are often combined with MSCs to provide tunable biocompatibility and enhanced cell functionality [202]. The addition of growth factors or gene transfer techniques to MSC-laden hydrogels has shown potential in further improving chondrogenesis. These innovative approaches aim to overcome limitations in current cartilage regeneration therapies and develop more effective MSC-based treatments for articular cartilage defects.

## 6. Challenges and Limitations

### 6.1. Cell Viability and Potency During Culture Expansion

MSCs show promise for cartilage regeneration, but their therapeutic potential can be affected by culture expansion. During expansion, MSCs tend to lose proliferation and differentiation capacity, particularly in osteogenesis and adipogenesis, although their chondrogenic potential is better preserved [203]. However, with extensive passaging, chondrogenic potency eventually declines, despite the retention of trophic repair properties [204]. To address these challenges, several strategies have been explored. For instance, microfluidic selection of medium-sized MSCs (17–21 μm) during expansion enhances proliferation and chondrogenic capacity [205]. Additionally, modulating WNT signaling by combining WNT3A with FGF2 during expansion preserves MSC chondrogenic potential and multipotency [203]. Furthermore, inhibiting WNT signaling during differentiation prevents hypertrophic maturation and calcification, thus addressing another obstacle in MSC-based cartilage repair [203]. Collectively, these approaches offer promising avenues for advancing MSC-based therapies for cartilage regeneration.

### 6.2. Scalability and Sourcing Issues

MSCs offer promising potential for cartilage repair but face considerable challenges in clinical applications. Although MSCs can differentiate into chondrocyte-like cells, they often produce fibrocartilage and may undergo hypertrophy, leading to mineralization and vascularization [206]. Successful cartilage regeneration requires the optimization of MSC sources, growth factors, and scaffolds to promote stable chondrogenesis while preventing hypertrophy [207]. Various MSC sources, including adipose tissue, bone marrow, and synovium, exhibit distinct biological activities and regenerative potentials [208]. Key challenges include ensuring the quality and durability of the repair tissue, achieving integration with surrounding host tissue, and preventing endochondral ossification [209]. Additionally, heterogeneity in MSC preparations and issues with quality control must be addressed. Despite these obstacles, MSC-based therapies remain a promising area of research, offering potential solutions to the limitations of current cartilage repair strategies [208,209].

### 6.3. Regulatory and Ethical Considerations

MSCs have shown promise in treating OA due to their regenerative and anti-inflammatory properties [210]. While both adipose-derived and bone marrow-derived MSCs have been studied, their effectiveness remains uncertain [211]. For example, a randomized controlled trial demonstrated that allogeneic bone marrow MSCs improve pain, function, and cartilage quality in patients with knee OA compared with hyaluronic acid injections [58]. However, a systematic review of 61 studies, including 14 high-quality trials, indicated inconsistency in MSC preparation methods and limited long-term follow-up data [212]. Despite these challenges, minimally manipulated adipose-derived MSCs have shown promising results in treating OA-related pain and improving joint function, with few complications observed in short- to mid-term follow-ups [210]. Further research is needed to establish standardized protocols and evaluate the long-term efficacy of these treatments.

### 6.4. Risks: Tumorigenicity, Immune Response, and Rejection

MSCs have shown promise in treating OA due to their immunomodulatory properties and ability to differentiate into chondrocytes [213]. While MSCs are generally considered safe, potential risks include tumorigenicity, unwanted immune responses, and transmission of adventitious agents [214,215]. The risk profile varies based on factors, such as cell type, differentiation status, administration route, and manipulation steps [214]. Although most small-scale clinical trials using MSCs have not reported major health concerns, some serious adverse events have occurred, emphasizing the need for continued research on their long-term safety [214]. Regulatory bodies have addressed the potential tumorigenicity of MSCs through quality control measures and nonclinical and clinical assessments [216]. Despite these risks, MSCs remain a promising therapeutic option for OA, offering regenerative potential and beneficial immunomodulatory effects [213].

### 6.5. How to Overcome MSCs Heterogeneity

MSCs exhibit significant heterogeneity, which poses challenges for their clinical applications [101,217]. This heterogeneity stems from various factors, including tissue sources, donor attributes, and manufacturing protocols [217]. To address this issue, researchers have explored marker-based isolation strategies to purify MSC subpopulations [101,218]. Additionally, the generation of induced pluripotent stem cell-derived MSCs (iMSCs) has emerged as a promising approach to reduce age- and tissue-related heterogeneity through epigenetic rejuvenation [219]. Meta-analyses of transcriptome data have revealed markers and biological processes characterizing MSC heterogeneity across various tissues [219]. Overcoming MSC heterogeneity is crucial for standardized production and reliable clinical applications, with strategies such as MSC pooling and iMSC generation showing potential [52,217,219].

### 6.6. How to Overcome Variability in Isolation and Expansion Protocols for MSCs

Variability in MSC isolation and expansion protocols can be addressed through standardization and optimization. Todtenhaupt et al. developed a robust method for umbilical cord-derived MSCs, demonstrating consistency across 90 donors [105]. Rojewski et al. successfully translated a standardized protocol for bone marrow-derived MSCs from validation to clinical manufacturing [106]. Alonso-Camino & Mirsch emphasized the importance of uniform protocols for MSCs from different sources to improve result comparability and clinical translation [220]. They proposed a standardized cGMP production method using xenogeneic-free medium. Detela et al. characterized growth kinetics of bone marrow-derived MSCs from multiple donors, identifying an operating window between passages 1–3 [221]. They suggested that early proliferation indicators could predict overall expansion potential. These studies collectively demonstrate that implementing standardized, optimized protocols can help overcome variability in MSC isolation and expansion, enhancing reproducibility and facilitating clinical applications.

### 6.7. How to Overcome Cell Viability After Delivery

MSCs show promise for treating various diseases, but poor post-transplantation viability remains a major challenge [222]. Several strategies have been explored to enhance MSC survival and function. Three-dimensional (3D) culturing of MSCs induces autophagy and suppresses ROS production, leading to improved cell viability [222]. Pharmacological preconditioning with celastrol, an antioxidant, significantly increases MSC survival and proangiogenic paracrine function when encapsulated in hydrogels [223]. Other approaches include priming with soluble factors, genetic modifications, and alternative cell delivery systems [224]. Despite these advancements, challenges persist in clinical translation, such as population heterogeneity, variability in isolation and expansion protocols, and safety concerns regarding teratogenic potential [99]. Ongoing research aims to address these issues and improve the efficacy of MSC-based therapies.

### 6.8. How to Overcome Cell’s Function After Transplantation

MSCs show promise in regenerative medicine, but their efficacy is limited by poor survival and function after transplantation. The hostile environment at transplantation sites, characterized by low oxygen and nutrients, negatively impacts MSC metabolism and survival [225]. Mitochondrial dysfunction, a primary cause of MSC death and senescence, is exacerbated in hyperglycemic conditions [226]. To enhance MSC transplantation outcomes, several strategies have been proposed. Melatonin, a pleiotropic molecule, can preserve MSC survival and function by reducing oxidative stress, promoting mitochondrial functionality, and facilitating mitochondrial transfer through tunneling nanotubes [227]. Preconditioning MSCs with physical, chemical, and biological factors can help maintain their stemness and improve their activities both in vitro and in vivo [228]. These approaches aim to overcome the challenges of maintaining MSC function after transplantation and enhance their therapeutic potential.

## 7. Conclusions

MSC therapy shows promise for treating OA and cartilage lesions. MSCs demonstrate chondrogenic and immunomodulatory potential, with AT-MSCs being particularly promising. Clinical studies have shown structural benefits, including cartilage repair, as evidenced by MRI and arthroscopy. MSCs can be administered through tissue engineering, scaffold-free injection, or cell-free exosome injection. The future outlook for MSC-based therapies in OA is optimistic, with ongoing research focusing on enhancing MSC regenerative potential through combinations with growth factors, biomaterials, and gene therapies. While most studies report clinical improvements and positive MRI findings, histological results are more controversial. Factors associated with better outcomes include younger age, smaller lesion size, and earlier OA stages. Despite promising results, high-quality clinical evidence remains limited, and no major adverse events have been reported. Furthermore, long-term safety and efficacy need to be confirmed through larger clinical trials. Further research is needed to optimize cell sources, manipulation techniques, and delivery methods for specific indications. Successfully addressing these challenges could unlock the full potential of MSC therapies, offering a regenerative, non-invasive solution for treating OA and mitigating age-related joint deterioration.

## Figures and Tables

**Figure 2 ijms-25-12911-f002:**
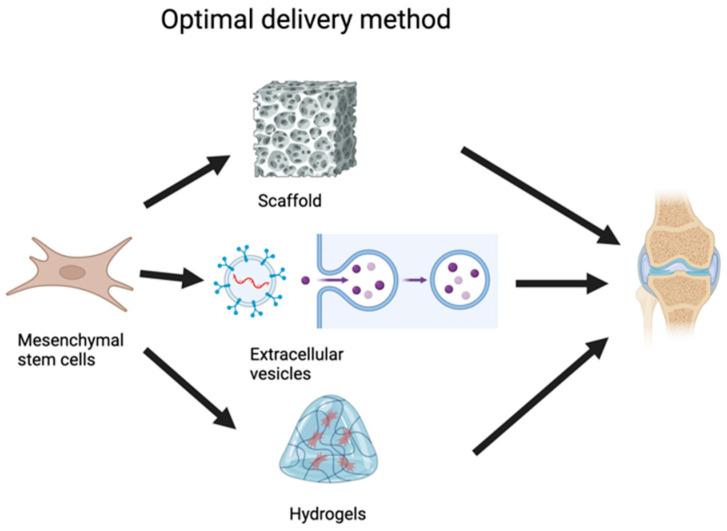
Optimal delivery methods for mesenchymal stromal cells (MSCs) in joint regeneration. The three main delivery methods are scaffold, extracellular vesicles, and hydrogels. Scaffolds are porous, supportive structures that enable MSCs to attach and proliferate, thereby maintaining cell viability and providing structural stability at the target site. Extracellular vesicles, which are cell-free particles derived from MSCs, contain signaling molecules (e.g., RNA, proteins) that facilitate tissue repair through intercellular communication, eliminating the need to introduce live cells. Hydrogels, gel-like materials that encapsulate MSCs, provide a hydrated environment that protects cells and enables their gradual release, supporting sustained therapeutic effects. These three delivery methods are directed toward the joint, representing their potential applications in regenerative treatments for conditions like osteoarthritis or cartilage damage. This figure highlights the adaptability of MSC-based therapies and the importance of selecting optimal delivery vehicles to enhance clinical outcomes.

**Table 1 ijms-25-12911-t001:** Search strategy outline.

Items	Specifications
Timeframe	From inception to 21 October 2024
Database	PubMed, Scopus, Web of Science, and Embase
Search terms used	“mesenchymal stromal cells”, “aging”, and “cartilage”
Inclusion and exclusion criteria	All references were SCI-indexed articles written in English.
Selection process	Two independent reviewers evaluated the titles and abstracts to determine eligibility.

**Table 2 ijms-25-12911-t002:** Status of MSC treatment for osteoarthritis registered at clinicaltrials.gov (n = 130).

Osteoarthritis	Status	Number
Open studies	Recruiting	18
	Enrolled by invitation	1
	Not yet recruiting	10
	Active, not recruiting	6
Closed studies	Completed	53
	Suspended	0
	Terminated	3
	Withdrawn	5
Unknown status		34

**Table 3 ijms-25-12911-t003:** Transplanted cell types and sources in clinical trials (n = 130).

Cell Types	Number	Percentage
ADSC	43	33.1
UCMSC	31	23.8
BMSC	36	27.7
Others	10	7.7
Exosomes	1	0.8
Unknown	9	6.9
Sources		
Autologous	71	54.6
Allologous	48	36.9
Unknown	11	8.5

ADSC, adipose-derived stromal cells; UCMSC, umbilical cord mesenchymal stromal cells; BMSC, bone marrow stromal cells.

## Data Availability

All relevant data are reported in the article.

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
