# Peer review of "Mesenchymal Stromal Cells for Aging Cartilage Regeneration: A Review"

_ijms, 2024, doi:10.3390/ijms252312911_

Round 1
Reviewer 1 Report
Comments and Suggestions for Authors
Minor remarks:
Prefer mesenchymal stromal cells throughout the document.
Line 39: specify per 100,000/year inhabitants.
The importance of glucose levels in the paracrine environment in maintaining MSC survival should be specified.
Clinical trials: refs 54 & 56 are old. Are there robust data on larger, more recent clinical trials (e.g. ADIPOA or others)?
Clarify the importance of chondroxia in preventing enchondral ossification.
Define EV at the beginning of paragraph 5.2.1.
The risk of “off-target” adverse effects must be specified for CRISPR/Cas9.
Clarify the value of MRI for sequential non-invasive clinical monitoring
It seems to me that intra-articular injection of MSC has only a transient, symptomatic and non-structural effect. Please precise
Author Response
Minor remarks:
Q1. Prefer mesenchymal stromal cells throughout the document.
Response 1: We thank the reviewer’s comment. We have revised it accordingly.
Q2. Line 39: specify per 100,000/year inhabitants.
Response 2: We thank the reviewer’s comment. We have revised “year inhabitants” to “persons”. (page 2, line 49)
Q3. The importance of glucose levels in the paracrine environment in maintaining MSC survival should be specified.
Response 3: We thank the reviewer’s comment. We have added a paragraph to illustrate the importance of glucose in the paracrine environment ​ in maintaining MSC survival. (Section 2.2.2, page 6, lines 248-259) The statements read as
“Glucose plays a crucial role in maintaining MSC survival and function in the paracrine environment. Studies have shown that glucose deprivation, more than oxygen shortage, severely compromises MSC viability and functional maturation [68,69]. Under near-anoxic conditions, MSCs rely almost exclusively on glucose for ATP production through anaerobic glycolysis and possess limited internal glucose reserves [70]. Glucose not only promotes MSC survival but also enhances their angiogenic potential [69]. To address this challenge, researchers have developed glucose-releasing scaffolds and enzyme-controlled, nutritive hydrogels that provide physiological glucose levels to MSCs [71]. These approaches have demonstrated improved MSC viability and paracrine functions both in vitro and in vivo. The findings highlight the importance of glucose in the paracrine environment for maintaining MSC survival and suggest that glucose supplementation strategies could enhance the efficacy of MSC-based therapies.”
Q4. Clinical trials: refs 54 & 56 are old. Are there robust data on larger, more recent clinical trials (e.g. ADIPOA or others)?
Response 4: We thank the reviewer’s comment. We have replaced ref 54 and 56 with the newer references.
Q5. Clarify the importance of chondroxia in preventing enchondral ossification.
Response 5: We thank the reviewer’s comment. We have added a paragraph to illustrate the chondroxia in preventing enchondral ossification. (section 3.2.2, pages 9-10, lines 399-410) The statements read as
“Endochondral ossification (EO) is a crucial process in bone formation and repair, involving the transformation of cartilage into bone [117]. Recent research challenges the traditional view that hypertrophic chondrocytes undergo apoptosis, suggesting instead that they may transdifferentiate into osteoblasts [117,118]. This chondrocyte-to-osteoblast transdifferentiation occurs through three models: direct transdifferentiation, dedifferentiation to redifferentiation, and chondrocyte to osteogenic precursor [118]. Epigenetic factors, including DNA methylation, histone modifications, and non-coding RNAs, play crucial roles in regulating EO and chondrogenesis [119]. Understanding these processes is essential for developing treatments for skeletal diseases and OA, which involves disruptions in chondrocyte homeostasis [120]. Maintaining healthy articular cartilage is vital for joint function and longevity, and further research into cartilage development and homeostasis is necessary for establishing regenerative therapies [120].”
Q6. Define EV at the beginning of paragraph 5.2.1.
Response 6: We thank the reviewer’s comment. We have added a paragraph to illustrate EV. (Section 5.2.1, page 11, lines 485-497) The statements read as
“EVs are small, membrane-enclosed structures released by cells into the extracellular space, ranging in size from 30 nm to 5 μm [140]. They play crucial roles in cell-to-cell communication by transferring functional biomolecules between cells [141]. EVs are classified into several subtypes based on their biogenesis, including exosomes (40-100 nm), microvesicles (100-1000 nm), and apoptotic bodies [142]. Recent studies have identified additional EV types such as autophagic EVs, stressed EVs, and matrix vesicles [142]. EVs are involved in various physiological and pathological processes, including tumor immunosuppression and metastasis [142]. They show promise as biomarkers, therapeutic agents, and drug delivery vehicles [143]. However, the cellular and molecular mechanisms governing EV functions are not fully understood, partly due to technical challenges in studying these small particles [143]. Despite these hurdles, the versatility and potential of EVs in regenerative medicine make them an exciting area of research for future therapeutic applications [144].”
Q7.The risk of “off-target” adverse effects must be specified for CRISPR/Cas9.
Response 7: We thank the reviewer’s comment. We have added a paragraph to illustrate off-target adverse effects. (Section 5.3, page12, lines 550-560) The statements read as
“CRISPR/Cas9 gene editing has shown promise for treating diseases, but off-target effects remain a significant concern [166,167]. These unintended alterations can include small indels, large deletions, and structural variations, posing risks to patients [167]. Various methods have been developed to detect and assess off-target effects, with ongoing efforts to enhance CRISPR precision [166]. Population-specific analysis of potential off-target sites is crucial, as polymorphisms can affect cleavage likelihood and create new PAM sequences [168]. Additionally, complex on-target outcomes, such as large deletions and gene rearrangements, have been observed and require careful consideration [169]. As CRISPR-based therapies advance towards clinical applications, comprehensive evaluation of both off-target and on-target effects is essential to ensure safety and efficacy in gene therapy development [167,169].”
Q8. Clarify the value of MRI for sequential non-invasive clinical monitoring
Response 8: We thank the reviewer’s comment. We have added a paragraph to illustrate the value of MRI for sequential non-invasive clinical monitoring. (Section 6.4, page 15, lines 664-677) The statements read as
“6.4 The value of magnetic resonance imaging for sequential non-invasive clinical monitoring
Magnetic resonance imaging (MRI) has emerged as a valuable tool for non-invasive monitoring of stem cell therapy in OA. Studies have shown that MRI can effectively evaluate cartilage regeneration and treatment outcomes [199]. T1 and T2 mapping techniques have been used to assess cartilage quality and correlate with clinical scores [199]. Pre-treatment MRI findings, such as cartilage lesions and bone marrow lesions, have been associated with improved clinical outcomes following adipose-derived stem cell therapy [200]. Advanced imaging techniques, including diffusion-weighted MRI and contrast-enhanced ultrasound, show promise for stem cell treatment evaluation [201]. In a study using Wharton's jelly mesenchymal stromal cells for knee OA, MRI scans demonstrated significant improvements in various parameters, including cartilage loss and bone marrow lesions, correlating with functional improvements [202]. These findings highlight the potential of MRI as a crucial tool for monitoring stem cell therapy efficacy in OA.”
Q9. It seems to me that intra-articular injection of MSC has only a transient, symptomatic and non-structural effect. Please precise
Response 9: We thank the reviewer’s comment. We have added a paragraph to illustrate the effect of intra-articular injection of MSC on cartilage regeneration. (Section 6.2, page 13, lines 595-605) The statements read as
“Recent studies have investigated the efficacy of intra-articular injections of MSCs for knee OA treatment. Clinical trials have shown improvements in pain, function, and quality of life following MSC injections [180–182]. Magnetic resonance imaging revealed decreased cartilage defects and increased cartilage volume in treated knees [180,181]. Histological analysis demonstrated hyaline-like cartilage regeneration [181]. Higher doses of MSCs (1.0 × 10^8 cells) generally produced better outcomes [180,181]. However, concerns about the durability of clinical and structural improvements beyond one year were noted, particularly for lower doses [180]. No serious adverse events were reported in these studies [180–182]. While promising, larger randomized clinical trials with long-term follow-up are needed to establish the efficacy and optimal dosing of MSC injections for knee OA[59].”
Reviewer 2 Report
Comments and Suggestions for Authors
Authors provide a review of the therapeutic potential of mesenchymal stem cells (MSCs) for cartilage regeneration in the context of aging and osteoarthritis (OA). They highlight the limitations of current treatments, which primarily target symptom relief rather than cartilage repair.
In its current form, the manuscript has significant potential but requires revisions to improve its depth, organization, and presentation. Addressing these issues will enhance its scholarly impact and make it more suitable for acceptance:
1. As a review article, its primary role is to synthesize existing knowledge, but currently it offers limited original insights or critical analyses. While challenges and solutions are outlined, they are often generic. A deeper critique of unresolved issues or innovative perspectives would enhance its value.
2. The search appears to be limited to PubMed, which might exclude studies indexed in other databases like Scopus, Web of Science, or Embase. This could lead to an incomplete representation of the available literature.
3. Please either include global statistics or clarify the relevance of the Taiwan data to ensure applicability to a broader audience.
4. Could you provide quantitative data comparing MSC yields, proliferative capacities, or differentiation efficiencies across different sources to substantiate their relative advantages?
5. Section 3.1.1 Animal models demonstrating MSC-induced cartilage regeneration - Variability in outcomes across different species is not addressed, making it difficult to assess the translatability of these results.
6. Table 2 - The formatting of the table could be clearer and more professional. Currently, it appears cluttered and difficult to read due to inconsistent spacing and abbreviations that may not be immediately understandable to readers.
7. Section 2.2 Mechanisms of Cartilage Regeneration - Much of the information in this section reiterates well-established knowledge about MSCs and their mechanisms without offering new perspectives or addressing current debates in the field.
8. Section 3.2.2 Challenges in translating preclinical success to clinical practice - should be expanded and analyzed in greater depth to provide a more comprehensive evaluation of the obstacles and potential solutions, including more research articles.
9. The section on future advancements should be expanded to include specific strategies for overcoming current limitations in MSC therapy, such as innovative approaches in cell engineering or personalized medicine.
Author Response
Reviewer 2
Authors provide a review of the therapeutic potential of mesenchymal stem cells (MSCs) for cartilage regeneration in the context of aging and osteoarthritis (OA). They highlight the limitations of current treatments, which primarily target symptom relief rather than cartilage repair.
In its current form, the manuscript has significant potential but requires revisions to improve its depth, organization, and presentation. Addressing these issues will enhance its scholarly impact and make it more suitable for acceptance:
Q1. As a review article, its primary role is to synthesize existing knowledge, but currently it offers limited original insights or critical analyses. While challenges and solutions are outlined, they are often generic. A deeper critique of unresolved issues or innovative perspectives would enhance its value.
Response 1: We thank the reviewer’s comment. We have added a deeper critique of unresolved issues or innovative perspectives (in the below points).
Q2. The search appears to be limited to PubMed, which might exclude studies indexed in other databases like Scopus, Web of Science, or Embase. This could lead to an incomplete representation of the available literature.
Response 2: We thank the reviewer’s comment. We have added Scopus, Web of Science, and Embase databases. (Table 1)
Q3. Please either include global statistics or clarify the relevance of the Taiwan data to ensure applicability to a broader audience.
Response 3: We thank the reviewer’s comment. We have added the global statistics in the first paragraph. (page 1, lines 38-48) The statements read as:
“OA is a major global health challenge, with increasing prevalence and burden worldwide. From 1990 to 2017, the global age-standardized prevalence of OA increased by 9.3% to 3,754.2 per 100,000 [2]. By 2019, there were approximately 527.8 million prevalent cases globally [3]. Hip and knee OA are particularly significant, ranking as the 11th highest contributor to global disability in 2010 [4]. The burden of OA is expected to continue rising due to population growth, aging, and increasing obesity rates [5]. Socio-demographic factors play a role, with a positive association between age-standardized incidence rates and the Socio-demographic Index [3]. To address this growing burden, it is crucial to implement measures targeting risk factors such as high body mass index and to improve awareness among populations and policymakers about OA management [2].”
Q4. Could you provide quantitative data comparing MSC yields, proliferative capacities, or differentiation efficiencies across different sources to substantiate their relative advantages?
Response 4: We thank the reviewer’s comment. We have added a paragraph to describe the requested items. (Section 2.1.3, page 4, lines 135-160) The statements read as:
“2.1.3 Comparing MSC yields, proliferative, and differentiation capacity from different sources
MSCs from different sources exhibit varying characteristics, impacting their potential for regenerative medicine applications. Studies comparing MSCs from BM, AT, UC, and decidua parietalis (DeP) reveal source-specific differences in yield, proliferation, and differentiation capacities. AT-MSCs demonstrated the highest isolation yield (BM-MSC-1 × 10^3 cells/mL of bone marrow aspirate, AT-MSC-2.5 × 10^6 cells/g of adipose tissue and AM-MSC-5.6 × 10^6 cells/g of amniotic tissue) and proliferation rates (absorbance at 572 nm after 240 hours, AT- 0.7, BM: 0.4, AT-0.2) [29], while UC-MSCs showed superior proliferation (doubling time: 17.7 vs. 21.9 hours) and colony formation (16% vs. 13.6%) compared to DeP-MSCs [30]. BM-MSCs exhibited better differentiation abilities, making them preferable for orthopedic applications [31]. Immunophenotyping confirmed typical MSC surface markers across sources, with slight variations in quantitative data between laboratories [32]. Functional assays revealed source-specific differences in angiogenic and immunomodulatory properties, with BM-MSCs enhancing tubulogenesis and AT-MSCs showing superior immunosuppressive abilities [32]. Dental pulp (DP)-MSCs demonstrate superior osteogenic differentiation potential and lower apoptosis rates compared to UC-MSCs, but UC-MSCs exhibit higher proliferation capacity [33]. While DP-derived cells show higher colony-forming efficiency, BM-MSCs have greater expansion success and differentiation potential [34]. UC-MSCs express higher levels of MSC surface markers like CD29, CD34, CD44, CD73, CD105, CD146, and CD166 compared to DP, although both sources exhibit similar overall MSC marker expression [35]. Gene expression patterns differ between UC and DP, with UC showing higher expression of cell proliferation and angiogenesis-related genes, while DP expresses more growth factor and signal transduction-related genes [35]. These findings highlight the importance of considering tissue origin when selecting MSCs for specific clinical applications.”
Q5. Section 3.1.1 Animal models demonstrating MSC-induced cartilage regeneration - Variability in outcomes across different species is not addressed, making it difficult to assess the translatability of these results.
Response 5: We thank the reviewer’s comment. We have added several paragraphs to illustrate the variability in outcomes across different species. (section 3.1.1, pages 6-7, lines 288-306) The statements read as:
“In various OA animal models, rats were commonly used, with studies reporting improved cartilage preservation and reduced damage after MSC injections [82,83]. In rabbits, most studies found improvements in cartilage repair and reduced osteoarthritis progression based on histological and radiological assessments [84]. Mice models showed promising results in fracture-induced and collagenase-induced OA [85]. Interestingly, a study on Guinea pigs with spontaneously developing OA reported beneficial effects of MSC treatment [86]. Larger animals provided mixed results. In sheep and goat models, some studies reported improvements, while others found no significant differences between MSC-treated and control groups [87,88]. A study using horses found no significant treatment effects on gross pathological observations [89]. Several studies using pigs found significant treatment effects on gross pathological observations [78,81]. Overall, while most studies across species reported some degree of improvement after MSC treatment, the outcomes varied considerably [90]. Smaller animals generally showed more consistent positive results, while larger animal models produced more mixed outcomes. However, it's important to note that the review found substantial heterogeneity among studies in terms of methodologies, MSC types, doses, and evaluation methods [82]. Additionally, the evidence quality for all outcomes was either low or very low, highlighting the need for further high-quality research before drawing definitive conclusions about the efficacy of MSC treatments across species [82].”
Q6. Table 2 - The formatting of the table could be clearer and more professional. Currently, it appears cluttered and difficult to read due to inconsistent spacing and abbreviations that may not be immediately understandable to readers.
Response 6: We thank the reviewer’s comment. We have revised the table format to make it clear to understand. (Table 2)
Q7. Section 2.2 Mechanisms of Cartilage Regeneration - Much of the information in this section reiterates well-established knowledge about MSCs and their mechanisms without offering new perspectives or addressing current debates in the field.
Response 7: We thank the reviewer’s comment. We have added several paragraphs to discuss lifespan, homing effect, autologous or allogeneic, and number of transplanted cells. (section 2.2.1, page 5, lines 196-229) The statements read as:
“One of the argued issues is lifespan of transplanted cells. A study found no transplanted cells stained after 2 months of treatment [50]. Other research indicates that MSCs have a limited lifespan [51]. Injecting MSCs in gel form into the articular cavity has demonstrated improved patient outcomes [52], though implanting MSCs directly into cartilage defects under arthroscopy has shown even greater benefits [53]. In an animal study, bone marrow-derived MSCs cultured and implanted with a collagen-hyaluronic acid scaffold significantly enhanced type II collagen [54]. The therapeutic effect of MSCs direct injections need further exploration.
Another issue is stem cell’s homing effect. The homing effect of MSCs to injured joints is a crucial factor in their therapeutic efficacy [21]. However, challenges remain in ensuring MSC retention and engraftment in cartilage tissue. To address this, researchers have explored various strategies, including cell surface modification and the use of nanoparticles for improved targeting and gene delivery [55]. Additionally, advanced biomaterials have been investigated to enhance MSC engraftment to cartilage and optimize cell dosage [55]. These approaches aim to improve the overall efficacy of MSC-based therapies for OA treatment.
MSC therapy shows promise for treating knee OA, with both autologous and allogeneic sources demonstrating potential benefits. Allogeneic MSCs have shown improvements in pain, function, and cartilage quality compared to hyaluronic acid injections [56]. While both sources appear safe, autologous MSCs may offer superior efficacy and safety profiles [57]. However, allogeneic MSCs provide logistical advantages and consistent product quality [58]. Despite these findings, current evidence is limited, and more high-quality randomized controlled trials comparing autologous and allogeneic MSCs are needed to establish definitive recommendations for treating knee OA [57,58].
Transplanted cell numbers is also matter. Cell doses ranging from 24 to 100 million have been investigated, with moderate doses (40 million) appearing optimal for efficacy and safety [59,60]. Lamo-Espinosa et al. reported that the same beneficial effects of MSC treatment with different cell dosages (10, 40 and 100 million cells) can persist for up to 4 years after a single injection [61]. Additionally, the use of adipose-derived stromal vascular fraction has shown potential in improving knee function and reducing pain, with higher cell numbers (an average of 45 million) correlating with better outcomes [62]. While these studies demonstrate the safety and efficacy of MSC-based treatments for knee osteoarthritis, larger-scale, long-term clinical trials are needed to further validate these findings [59].”
Q8. Section 3.2.2 Challenges in translating preclinical success to clinical practice - should be expanded and analyzed in greater depth to provide a more comprehensive evaluation of the obstacles and potential solutions, including more research articles.
Response 8: We thank the reviewer’s comment. We have extended Section 3.2.2. We have added issues regarding the heterogeneity of MSC populations, variability in isolation and expansion protocols, and concerns about cell viability and function after delivery. (pages 8-9, lines 339-388) The statements read as:
“MSCs exhibit significant heterogeneity in morphology, phenotype, and function, which poses challenges for their therapeutic applications [99,100]. This heterogeneity is influenced by microenvironmental factors, including mechanical stiffness, which can alter MSC gene expression and commitment [101]. To address this variability, researchers have explored marker-based isolation strategies and high-throughput approaches to identify and purify MSC subpopulations (Si Chen et al., 2024). Studies have shown that culturing MSCs on soft surfaces or inhibiting specific pathways can promote a more homogeneous population [101]. Interestingly, while individual MSC clones may display varying immunosuppressive capabilities, exposure to pro-inflammatory cytokines (licensing) can eliminate these functional differences, resulting in uniformly enhanced immunosuppressive activity mediated by factors such as nitric oxide and prostaglandin E2 [102].
Recent studies highlight the variability in MSC isolation and expansion protocols, emphasizing the need for standardization. Todtenhaupt et al. developed a robust method for human umbilical cord-derived MSCs, optimizing critical variables and demonstrating consistency across 90 donors [103]. Rojewski et al. successfully translated a standardized protocol for bone marrow-derived MSCs from validation to clinical manufacturing, showing stable performance characteristics despite variations in starting material [104]. Shaz et al. found that local manufacturing processes contribute to variability in MSC expansion, while growth media supplements affect gene expression and cell function [105]. Wright et al. optimized protocols for canine umbilical cord-derived MSCs, addressing the challenge of maintaining these cells in culture for extended periods [106]. Their method improved cellular adherence, colony-forming efficiency, and population doubling times. These studies collectively emphasize the importance of developing standardized, robust protocols to enhance reliability and comparability of results across different donors and studies.
MSCs show promise for cellular therapies, but maintaining their viability during storage and transplantation is crucial. Cell viability should be at least 80% for clinical use [107]. However, MSC viability decreases rapidly after dissociation from culture dishes [108]. Storage solutions significantly affect cell survival, with viability dropping below 70% after 6 hours in common parenteral solutions [109]. Factors influencing MSC viability include cell density, dimethylsulfoxide (DMSO) concentration, and needle gauge. Maintaining cell density below 2 × 10^7 cells/mL and DMSO concentration below 0.5% can help preserve viability above 82% when using 25- or 27-gauge needles [110]. Various analytical techniques, such as membrane integrity assays, morphological studies, and fluorescence biosensors, are used to assess MSC viability [107]. Optimizing storage conditions and transplantation methods is crucial for maintaining MSC viability and therapeutic potential.
MSCs show promise in improving outcomes after organ transplantation. Studies have demonstrated the safety of MSC infusion after liver transplantation [111] and kidney transplantation [112]. MSCs have shown potential in treating poor graft function following hematopoietic cell transplantation, with improved hematological responses and reduced transfusion requirements [113]. However, the therapeutic effects of MSCs may be limited due to impaired function after infusion. Preconditioning methods are being explored to enhance MSC efficacy in kidney transplantation [114]. While MSC therapy appears safe and feasible, its benefits in organ transplantation are not yet fully established. Further research is needed to optimize MSC function post-transplantation and to demonstrate their potential advantages over standard immunosuppressive regimens. Larger prospective studies are required to confirm the efficacy of MSC therapy in transplantation settings [111,112].”
Q9. The section on future advancements should be expanded to include specific strategies for overcoming current limitations in MSC therapy, such as innovative approaches in cell engineering or personalized medicine.
Response 9: We thank the reviewer’s comment. We have added overcoming current limitations in MSC therapy mentioned in the previous comment. (Sections 6.4-6.10, pages 15-17) The statements read as:
“​​6.4 The value of magnetic resonance imaging for sequential non-invasive clinical monitoring
Magnetic resonance imaging (MRI) has emerged as a valuable tool for non-invasive monitoring of stem cell therapy in OA. Studies have shown that MRI can effectively evaluate cartilage regeneration and treatment outcomes [199]. T1 and T2 mapping techniques have been used to assess cartilage quality and correlate with clinical scores [199]. Pre-treatment MRI findings, such as cartilage lesions and bone marrow lesions, have been associated with improved clinical outcomes following adipose-derived stem cell therapy [200]. Advanced imaging techniques, including diffusion-weighted MRI and contrast-enhanced ultrasound, show promise for stem cell treatment evaluation [201]. In a study using Wharton's jelly mesenchymal stromal cells for knee OA, MRI scans demonstrated significant improvements in various parameters, including cartilage loss and bone marrow lesions, correlating with functional improvements [202]. These findings highlight the potential of MRI as a crucial tool for monitoring stem cell therapy efficacy in OA.
6.5 How to overcome MSCs heterogeneity
MSCs exhibit significant heterogeneity, which poses challenges for their clinical applications [99,203]. This heterogeneity stems from various factors, including tissue sources, donor attributes, and manufacturing protocols [203]. To address this issue, researchers have explored marker-based isolation strategies to purify MSC subpopulations [99,204]. Additionally, the generation of induced pluripotent stem cell-derived MSCs (iMSCs) has emerged as a promising approach to reduce age- and tissue-related heterogeneity through epigenetic rejuvenation [205]. Meta-analyses of transcriptome data have revealed markers and biological processes characterizing MSC heterogeneity across various tissues [205]. Overcoming MSC heterogeneity is crucial for standardized production and reliable clinical applications, with strategies such as MSC pooling and iMSC generation showing potential [50,203,205].
6.6 How to overcome variability in isolation and expansion protocols for MSCs
Variability in MSC isolation and expansion protocols can be addressed through standardization and optimization. Todtenhaupt et al. developed a robust method for umbilical cord-derived MSCs, demonstrating consistency across 90 donors [103]. Rojewski et al. successfully translated a standardized protocol for bone marrow-derived MSCs from validation to clinical manufacturing [104]. Alonso-Camino & Mirsch emphasized the importance of uniform protocols for MSCs from different sources to improve result comparability and clinical translation [206]. They proposed a standardized cGMP production method using xenogeneic-free medium. Detela et al. characterized growth kinetics of bone marrow-derived MSCs from multiple donors, identifying an operating window between passages 1-3 [207]. They suggested that early proliferation indicators could predict overall expansion potential. These studies collectively demonstrate that implementing standardized, optimized protocols can help overcome variability in MSC isolation and expansion, enhancing reproducibility and facilitating clinical applications.
6.7 How to overcome cell viability after delivery
MSCs show promise for treating various diseases, but poor post-transplantation viability remains a major challenge [208]. Several strategies have been explored to enhance MSC survival and function. Three-dimensional (3D) culturing of MSCs induces autophagy and suppresses ROS production, leading to improved cell viability [208]. Pharmacological preconditioning with celastrol, an antioxidant, significantly increases MSC survival and proangiogenic paracrine function when encapsulated in hydrogels [209]. Other approaches include priming with soluble factors, genetic modifications, and alternative cell delivery systems [210]. Despite these advancements, challenges persist in clinical translation, such as population heterogeneity, variability in isolation and expansion protocols, and safety concerns regarding teratogenic potential [97]. Ongoing research aims to address these issues and improve the efficacy of MSC-based therapies.
6.8 How to overcome cell’s function after transplantation
MSCs show promise in regenerative medicine, but their efficacy is limited by poor survival and function after transplantation. The hostile environment at transplantation sites, characterized by low oxygen and nutrients, negatively impacts MSC metabolism and survival [211]. Mitochondrial dysfunction, a primary cause of MSC death and senescence, is exacerbated in hyperglycemic conditions [212]. To enhance MSC transplantation outcomes, several strategies have been proposed. Melatonin, a pleiotropic molecule, can preserve MSC survival and function by reducing oxidative stress, promoting mitochondrial functionality, and facilitating mitochondrial transfer through tunneling nanotubes [213]. Preconditioning MSCs with physical, chemical, and biological factors can help maintain their stemness and improve their activities both in vitro and in vivo [214]. These approaches aim to overcome the challenges of maintaining MSC function after transplantation and enhance their therapeutic potential.
6.9 Potential for personalized regenerative medicine approaches
Personalized regenerative medicine using MSCs for OA holds significant potential because it provides treatments tailored to individual’s needs [215]. MSCs have a distinctive ability to regenerate damaged cartilage, modulate inflammation, and adapt to the specific biological environment of a patient’s joint [67]. By analyzing patient-specific factors, such as the severity of cartilage damage, inflammation levels, and genetic predispositions, treatments can be personalized to enhance efficacy [216]. Biomarkers play a crucial role in identifying early-stage OA and predicting disease progression, enabling more effective personalized treatments [217]. The concept of endotypes and phenotypes in OA helps explain variations in clinical manifestations, etiology, and pathophysiology, potentially improving patient selection for regenerative therapies [218]. While regenerative medicine initially focused on cell engraftment and differentiation, recent studies suggest that injected cells may primarily exert transient paracrine effects before undergoing apoptosis [219]. To justify the high cost of cell therapy, long-term cell survival and durable structural improvements are essential [219,220]. Understanding OA phenotypes and endotypes can contribute to more effective patient selection for regenerative treatments, potentially reducing healthcare costs and improving outcomes [218]. This personalized approach enables for optimization of MSC source (whether autologous or allogeneic), dosage, and potential combination with other therapies, such as growth factors or drugs, ensuring that the therapy is best suited to the individual’s condition [221,222]. As research advances, personalized MSC therapies could transform OA management by providing more effective, long-lasting, and targeted treatments, potentially reducing the need for invasive procedures like joint replacement [92].
6.10 Innovative approaches in cell engineering
Recent advances in MSC engineering for cartilage regeneration focus on enhancing chondrogenic differentiation and addressing hypertrophy. Three-dimensional cell sheet technology has emerged as a promising approach for fabricating hyaline-like cartilage constructs [223]. Multilayering MSC sheets can increase construct thickness and enhance cellular interactions, although an optimal thickness threshold exists for maximizing chondrogenesis [223]. Various strategies to improve MSC chondrogenesis include optimizing bioactive factors, culture conditions, and physical stimulation [157]. Notably, N-cadherin mimetic hydrogels have been shown to attenuate hypertrophy and enhance chondrogenesis by regulating cell metabolism, specifically glycolysis and fatty acid oxidation [224]. Mechanobiological stimulation of stem cells has also been investigated to recreate specific environmental conditions for chondrogenesis [225]. Hydrogels, both natural and synthetic, are often combined with MSCs to provide tunable biocompatibility and enhanced cell functionality [226]. The addition of growth factors or gene transfer techniques to MSC-laden hydrogels has shown potential in further improving chondrogenesis. These innovative approaches aim to overcome limitations in current cartilage regeneration therapies and develop more effective MSC-based treatments for articular cartilage defects.”
Round 2
Reviewer 2 Report
Comments and Suggestions for Authors
The authors have thoroughly and thoughtfully addressed all my comments in great depth. I sincerely appreciate their effort in considering my suggestions and incorporating them into the manuscript. I am pleased to recommend the acceptance of this manuscript in its current form, as it provides a valuable contribution to the field of stem cell research.
Author Response
Comment 1. The authors have thoroughly and thoughtfully addressed all my comments in great depth. I sincerely appreciate their effort in considering my suggestions and incorporating them into the manuscript. I am pleased to recommend the acceptance of this manuscript in its current form, as it provides a valuable contribution to the field of stem cell research.
Response 1: We thank the reviewer's comment.